# Extremality as a consistency condition on subregion duality

**Ronak M. Soni**

Department of Applied Mathematics and Theoretical Physics, University of Cambridge,
Wilberforce Road, Cambridge, CB3 0WA, United Kingdom

ronakmsoni@gmail.com

## Abstract

In JT gravity coupled to a CFT, I argue without using the path integral that the entanglement wedge of a boundary region is bounded by a quantum extremal surface (QES). For any candidate not bounded by a QES, a unitary in the complement can make reconstruction within the candidate inconsistent with boundary causality. The case without islands is a direct consequence of subregion duality, and the case with islands can also be dealt with with a stronger assumption.

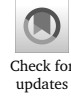

# 1 Introduction

In QFT, it is easy to define a local subsystem: it is merely the algebra of operators of the causal development of a partial Cauchy slice. The situation changes drastically when we couple a QFT to gravity, even in the extreme weak-coupling limit; see [1, 2] for example. This is because gravity has diffeomorphism gauge-invariance. At a first pass, the issue is that it is hard to define a subregion in a diffeomorphism-invariant way. However, one can define a subregion using appropriate dressings. The main issue is not so much defining a *subregion* as defining a *subsystem*: what we really need to worry about is whether the degrees of freedom in the subregion can be considered as independent from the complement in any sense, given the gauge constraints.

There are two main problems with understanding whether any given subregion is a subsystem. The first is that we define local operators using gravitational dressing, and the number of possible gravitational dressings diverges with human creativity.[1]

The second regards the meaning of a subsystem. Unlike QFT, we cannot define a subsystem to be an algebra of local operators, since the product of enough local operators can change the structure of spacetime; a true algebra has to be an algebra in a UV-complete theory, whereas we are interested in subsystems in low-energy effective theory. There have been many suggestions to deal with this problem, see especially [5–8]; I use the phrase 'approximate subalgebra' in the introduction to avoid getting into the weeds. These two choices are intertwined, since a dressing may make sense for only some approximate subalgebras and not others.[2]

Thus, to make progress on the question of whether a gravitational subregion contains a gravitational subsystem, we need an approach immune to clever interlocutors who can find new dressings and approximate subalgebras that are not imagined in our philosophies.[3] Such an approach is given by consistency conditions, conditions that should be true whenever there is a subsystem. Non-violation of these conditions may be inconclusive, but violation of a consistency condition is a sure sign of inconsistency.

Entanglement wedge reconstruction [6, 10–13] is one specific situation where it is possible to write down such a consistency condition. A boundary subregion $B$ encodes the EFT of a subregion $W_E[B; \Psi]$ of the bulk, meaning that every operator in the latter can be represented as one in the former. Note that the bulk region depends also on the boundary state $\Psi$. $W_E[B; \Psi]$, called the entanglement wedge (EW), is the region between $B$ and the minimal quantum extremal surface (QES) $X \sim B$ homologous to it [14], in the metric $g_\Psi$ dual to $\Psi$. When a bulk subregion is bounded by a QES, I will call the region itself extremal, for conciseness. Quite non-trivially, the entanglement wedge is also the subregion dual to $B$, in the sense that bulk EFT operators in $W_E[B; \Psi]$ can be reconstructed in $B$.

---

[1]To be more accurate and less hyperbolic, we do not have a classification of all types of dressings. See [3,4] for a recent example of a novel type of dressing.

[2]For example, an operator defined to be at fixed distance $\ell$ from the boundary along some geodesic implicitly contains a projector onto states for which the geodesic has length at least $\ell$. This projector might not play well with locality of the subalgebra, and because of the non-linearity of gravity the importance of this effect can depend on what order of $G_N$ we are working at.

[3]One approach to such a question was suggested recently in [9]; my approach is different.

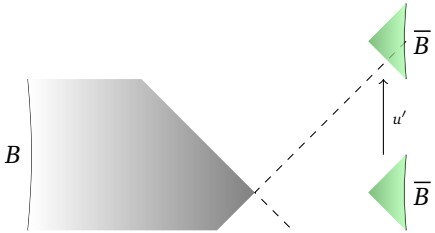

Figure 1: The strategy. For $b$ not extremal, it is possible to push the causal wedge of $\overline{B}$ (green) outside $\overline{b}$, using a unitary $u'$ localised in $\overline{b}$. This means that $b$ is not a valid candidate for the entanglement wedge of $B$.

In this work, I provide a new perspective on (a weaker version of) this non-trivial statement: extremality of the bulk subregion follows from consistency of subregion duality with causality in the boundary theory. I first argue that subregion duality implies a condition that I call "complementary causal wedge exclusion" (CCWE), and then that CCWE can be violated for any putative bulk dual of $B$ that is not extremal.

Suppose someone claims that $b$ is dual to $B$. By boundary causality, since $B$ is spacelike to $\overline{B}$, operators in $b$ should commute with those in $\overline{B}$; and further, the action of a unitary in $\overline{b}$ should not change this. In other words, it is not possible to engineer a situation like that in figure 1 if the shaded region is the entanglement wedge. This is the CCWE condition. It can be thought of as a dual of the famous causal wedge inclusion (CWI) condition [10,15].

[16–18] effectively showed that CCWE is violated for the causal wedge.[4] This work is a generalisation of the above: CCWE is violated unless $W_E[B; \Psi]$ is bounded by a QES, in JT gravity coupled to a 2d CFT. Thus, extremality of the EW is a more-or-less direct consequence of subregion duality.

Previous proofs of extremality of the EW include replica trick proofs [11,12,19], a proof from shape deformations of the modular Hamiltonian [20], and (in the special case of JT gravity) an operator algebraic proof [21–23]. This consistency condition approach provides a new perspective compared to the above, since it implies the subregion duality can, *in principle*, hold *only* for extremal wedges. It should be noted also that extremality of the EW has been shown to pass a number of important consistency conditions from subregion duality ever since the EW was defined, see [10,14] and follow-ups. CCWE goes beyond these consistency conditions in an important way: *only* extremal regions satisfy it.

Another reason this result is interesting is the connection between quantum gravity and quantum information. Assuming that holography has a quantum error-correcting structure is enough to derive a QES-like formula, with the role of the area being played by a state-independent operator[5] in the modular Hamiltonian [6,8,24–27]. However, a concrete identification of this state-independent operator as the geometric object 'area of the QES,' and the code subsystem as the region bounded by the minimal QES comes indirectly via the replica trick. The analysis here shows, in the cases where it applies, that extremal wedges are the only consistent choice of code subsystem, allowing for a direct application of these error-correction results to holography.

## Plan

Section 2 introduces the main new idea of this paper, the 'complementary causal wedge exclusion' consistency condition. The rest of the paper applies this condition to prove (quantum) extremality of the entanglement wedge, in JT gravity coupled to a CFT.

---

[4]More precisely, the complement of the causal wedge of the complementary boundary region, $\overline{W_C[\overline{B}; \Psi]}$.

[5]More precisely, it is state-independent at first subleading order in $G_N$. Quantum extremisation makes it a 'little bit' state-dependent.

In section 3, I introduce JT gravity and the Connes cocycle flow in 2d CFT. Section 4 argues that CCWE implies extremality under the assumption that the EW has a single connected component. This argument is a straightforward extension of that in [17]. Section 5 extends this to include the possibility of islands, under some assumptions.

I end with some discussion in section 6.

### Notation

1. An overline over a region $b$, $\overline{b}$, denotes the set of points spacelike-separated from $b$.

2. A causally complete region satisfies $b = \overline{\overline{b}}$.

3. The codimension-two boundary of a bulk region $b$ is $\eth b$. We have $\eth b = \eth \overline{b}$.

4. Small letters denote bulk regions/algebra/operators/states and capital letters denote boundary regions/algebras/operators/states. One exception to this is Hilbert spaces, where all Hilbert spaces are $\mathscr{H}$ with different subscripts; $\mathscr{H}_{bd}$ is the boundary Hilbert space and the bulk Hilbert space is $\mathscr{H}_{bulk}$.[6]

   A calligraphic small letter $\mathscr{a}$ refers to a QFT subalgebra in the bulk, $\mathscr{a} \in B(\mathscr{H}_{bulk})$.

   A particularly important, and potentially confusing, example of this notation is the Connes cocycle flow unitary $u'$ that will be my main workhorse: it is an operator in the bulk EFT, and does not (a priori) have anything to do with a similar unitary in the boundary theory.

5. A boundary state $|\Psi\rangle$ is dual to a bulk metric $g_\Psi$ and a bulk QFT state $|\psi\rangle \in \mathscr{H}_{g_\Psi}$. In JT gravity, the metric is independent of $\Psi$, and so the Hilbert space is just $\mathscr{H}_{bulk}$, but I include the dilaton configuration in $g_\Psi$.

6. For a bulk operator $a$, I denote by $a|\psi\rangle$ a state in $\mathscr{H}_{bulk}$, i.e. no backreaction is taken into account. The state *with* the backreaction taken into account I will denote by $a|\Psi\rangle$; the reader should think of this as bad notation for "embed the bulk operator $a$ into the boundary algebra $B(\mathscr{H}_{bd})$ and then act on the boundary state $|\Psi\rangle$."

## 2 Complementary causal wedge exclusion

Let me now introduce the consistency condition that non-extremal wedges will turn out to violate. I call it "complementary causal wedge exclusion" (CCWE); it is heuristically the Haag dual of causal wedge inclusion (CWI) [10, 15].

Divide the boundary CFT into two complementary subregions $B, \overline{B}$ with algebras $\mathscr{A}, \mathscr{A}'$; allow $B, \overline{B}$ to include reference systems so that the state $|\Psi\rangle$ on $B \cup \overline{B}$ is pure. Without loss of generality, we assume that the CFT is not coupled to any such reference systems.[7] To each of $B, \overline{B}$ we can associate a bulk causal wedge $W_C[B; g_\Psi], W_C[\overline{B}; g_\Psi]$, where $g_\Psi$ denotes the spacetime dual to $\Psi$.

---

[6]In most gravitational theories, we would need to define one Hilbert space for every on-shell bulk metric. JT gravity simplifies matters for us, however.

[7]We can always take the state on $B \cup \overline{B}$ at a given time and evolve it to all times with a decoupled, time-independent Hamiltonian; in the bulk, this gives us a spacetime with the usual AdS boundary conditions.

If coupling is allowed, CWI becomes a more involved statement [16], and so does CCWE. Further, the technique in section 3 does not straightforwardly generalise to deriving a contradiction with this more careful statement. However, this is not a genuine restriction, since we are interested in states rather than processes.

Assuming subregion duality is true, we can also associate to $B$ a bulk dual $W_E[B;\Psi]$. It is the largest bulk region such that for 'simple' local unitaries $u \in \mathscr{a}_{W_E[B;\Psi]}$,

$$\exists U \in \mathscr{A}, \qquad \text{s.t.} \qquad \forall u' \in \mathscr{a}'_{W_E[B;\Psi]}, \quad uu'|\Psi\rangle \approx Uu'|\Psi\rangle. \tag{1}$$

The precise statement, with a characterisation of the class of $u$ allowed, can be found in [28]. What is important for us is that this class of $u$ is expected to include low-energy local bulk unitaries. Note that there is no locality restriction on $u'$, except that its support does not extend into $W_E[B;\Psi]$. This is the definition of the max-EW (the largest region that can always be reconstructed), meaning that by this definition complementary recovery — "$\overline{W_E[B;\Psi]} = W_E[\overline{B};\Psi]$" — is not guaranteed.

Causal wedge inclusion (CWI) implies that

$$CWI \implies W_E[B;\Psi] \supseteq W_C[B;g_\Psi], \qquad \text{and} \qquad \overline{W_E[B;\Psi]} \supseteq W_C[\overline{B};g_\Psi]. \tag{2}$$

CWI follows from HKLL reconstruction [29] or the timelike-tube theorem [30–33].[8] Complementary causal wedge exclusion (CCWE) is defined to be

$$CCWE: \quad \forall \text{ unitaries } u' \in \mathscr{a}_{\overline{W_E[B;\Psi]}}, \qquad W_E[B;\Psi] \subseteq W_E[B;u'\Psi] \subseteq \overline{W_C[\overline{B};u'\Psi]}. \tag{3}$$

The first inclusion means that $W_E[B;u'\Psi]$ contains a region with the same metric and quantum state of bulk fields as $W_E[B;\Psi]$. CCWE is the same as (1), assuming that there are reconstructible local unitaries everywhere in $W_E[B;\Psi]$.

It is possible for example that, in the spacetime $g_{u'\Psi}$, a region $r \subset W_E[B;\Psi]$ is causally connected to $W_C[\overline{B}, u'\Psi]$ but that none of the operators with support in $r$ can be reconstructed in $B$. This possibility arises from $W_E[B;\Psi]$ being a *largest* region satisfying a constraint, and that the class of unitaries that must be reconstructible are not specified in the bulk, continuum language. However, since $W_E[B;\Psi]$ is also the region in which *no* unitaries are reconstructible in the complement $\overline{B}$, this would mean that unitaries with support in $r$ are not reconstructible in *either* $B$ or $\overline{B}$. This could be possible, logically, if for some reasons all gauge-invariant unitaries with support in $r$ must have support in both $W_E[B;\Psi] \setminus r$ as well as $\overline{W_E[B;\Psi]}$; then, all unitaries with support in $r$ can only be reconstructed in the full boundary theory. It is reasonable to expect, however, that all low-energy local operators in $W_E[B;\Psi]$ can be reconstructed, invalidating this scenario. To deal with this subtlety, I will take CCWE to be *assumption* A.1.

# 3 JT gravity coupled to a 2d CFT

## 3.1 The theory

The action of JT gravity [35–40] minimally coupled to a CFT is

$$I_{\text{JT}} = \chi S_0 + \frac{1}{4\pi} \int_{\mathscr{M}} \phi(R+2) + \frac{1}{2\pi} \int_{\partial\mathscr{M}} \phi(K-1) + I_{\text{CFT}}[g], \tag{4}$$

where $\chi$ is the Euler characteristic of $\mathscr{M}$. I have set the AdS length scale to 1. The boundary condition is

$$\phi\big|_{\partial\mathscr{M}} = \frac{\phi_r}{\epsilon}. \tag{5}$$

$1/\phi_r$ plays the role of $G_N$ in this theory, in that it controls the backreaction effects. We take the bulk matter to be a CFT with central charge $c$ satisfying

$$1 \ll c \ll \phi_{\text{r}}. \tag{6}$$

---

[8]The equivalence was first pointed out in [34].

Further, I also assume that the bulk states of interest are such that stress tensor fluctuations are $\mathcal{O}(1)$ and entropies are $\mathcal{O}(c)$. These assumptions allow us to ignore fluctuations in the dilaton $\phi$ while being sensitive to quantum effects in the bulk matter.

The equation of motion from the variation of the metric, after a convenient trace-reversal, is

$$\boldsymbol{\nabla}_i \partial_j \phi - g_{ij}\phi + 2\pi \langle t_{ij}\rangle = 0 , \tag{7}$$

where $t_{ij}$ is the stress tensor of the 2d CFT; it is a small letter because it is an operator in the bulk theory. Here, conformal invariance and a renormalisation of $S_0$ have been used to set the trace $\langle t_i^i \rangle = 0$. For general matter, we should replace $t_{ij} \to t_{ij} - g_{ij}t_k^k$. Due to the dilaton equation of motion, all solutions are locally AdS$_2$. Different geometries correspond to different dilaton profiles; by the boundary condition (5), this can be thought of also as different (conformal) trajectories for the two boundaries.

Since we are only interested in one topology, the metric can be gauge-fixed to

$$ds^2 = \frac{4 dw d\bar{w}}{(1 - w\bar{w})^2} . \tag{8}$$

$\bar{w}$ increases towards the past. There is a residual $SL(2, \mathbb{C})$ gauge freedom because of the isometries of this metric, which can be used to place any point at the coordinate location $w = \bar{w} = 0$.

A boundary state $|\Psi\rangle$ is dual to a dilaton profile $\phi(w, \bar{w})$, which I also call $g_\Psi$, and a bulk state $|\psi\rangle$. This boundary state is a state in two nCFTs, which we name $\overline{B}$ and $B$.

## 3.2 Connes cocyle flow and the QHANEC in a 2d CFT

Consider a 2d CFT of central charge $c$ on a Lorentzian manifold $\mathcal{M}$ with metric

$$ds^2 = e^{2\Omega}\widehat{ds}^2 , \qquad \widehat{ds}^2 = dw d\bar{w} . \tag{9}$$

The Weyl factor is $\Omega = \log[2/(1 - w\bar{w})]$ for AdS$_2$. The subregion of interest $\overline{b}$ is a wedge extending to the right boundary,

$$\overline{b} = \{w > w_0, \bar{w} > \bar{w}_0\} . \tag{10}$$

Assume that $w, \bar{w}$ extend to $\infty$; I will argue that this is the case of interest later. Define $\delta b \equiv \partial b \setminus \partial \mathcal{M}$ as the part of the boundary of $b$ that is not on the boundary of the spacetime.

There is a special state in the CFT, the vacuum $|\omega\rangle_g$, which is the Weyl transformation of the vacuum $|\omega\rangle_{\widehat{g}}$ on the metric $\widehat{g}$.[9] Denoting by $t_{ij}, \widehat{t}_{ij}$ the stress tensor operators on the metrics $g, \widehat{g}$ respectively, they are related as

$$t_{ij} = \widehat{t}_{ij} + f_{ij}(c, \Omega, \widehat{g})\mathbb{1} . \tag{11}$$

An explicit expression for $f$ can be found in e.g. [41]; the important property is that it is a c-number.

The vacuum state $\omega$ has a local modular Hamiltonian for the region $\overline{b}$. The (one-sided) modular Hamiltonian is the log of the reduced density matrix,

$$k_{\omega;\overline{b}} = -\log \rho_{\omega;\overline{b}} . \tag{12}$$

This is represented by a small letter since it is an operator in the bulk CFT. The subtracted (one-sided) modular Hamiltonian is

$$\Delta k_{\omega;\overline{b}} = k_{\omega;\overline{b}} - \langle \omega | k_{\omega;\overline{b}} | \omega \rangle = 2\pi \int_{w_0}^{\infty} dw (w - w_0)\widehat{t}(w) , \tag{13}$$

---

[9]The boundary condition at $\partial \mathcal{M}$ won't be important, except that it is conformally invariant.

where $\widehat{t}(w) = \widehat{t}_{ww}(w)$ is the stress tensor on the metric $\widehat{g}$. The error from using the stress tensor on the wrong metric is a c-number that cancels out in the subtraction.

The relative entropy is defined as

$$S_{\mathrm{rel}}(\psi|\omega;\overline{b}) = \left\langle \psi \left| k_{\omega;\overline{b}} - k_{\psi;\overline{b}} \right| \psi \right\rangle \geq 0. \tag{14}$$

Adding and subtracting $\langle \omega | k_{\omega;\overline{b}} | \omega \rangle$, it can be rewritten as

$$S_{\mathrm{rel}} = \left\langle \psi \left| \Delta k_{\omega;\overline{b}} \right| \psi \right\rangle - \Delta S_{E,\psi}, \qquad \Delta S_{E,\psi} = S_E(\psi;\overline{b}) - S_E(\omega;\overline{b}). \tag{15}$$

The first term can be written using (13) as

$$\left\langle \Delta k_{\omega;\overline{b}} \right\rangle_\psi = 2\pi \int_{w_0}^{\infty} \mathrm{d}w \, (w - w_0) \langle \Delta t(w) \rangle_\psi, \qquad \Delta t(w) \equiv t(w) - \langle t(w) \rangle_\omega. \tag{16}$$

I have used the standard notation that $\langle O \rangle_\psi = \langle \psi | O | \psi \rangle$. Relative entropy is monotonic; for any system $c \subseteq \overline{b}$, $S_{\mathrm{rel}}(\psi|\omega;c) \leq S_{\mathrm{rel}}(\psi|\omega;\overline{b})$. Using the form (13) for the modular Hamiltonian, monotonicity can be recast as the quantum half-averaged null energy condition (QHANEC) [42],

$$E_{QHA} \equiv -\partial_{w_0} S_{\mathrm{rel}}(\psi|\omega) = 2\pi \int_{w_0}^{\infty} \mathrm{d}w \, \langle \Delta t(w) \rangle + \partial \Delta S_{E,\psi}(w_0) \geq 0. \tag{17}$$

The inequality follows from the fact that increasing $w_0$ makes the region smaller. I will drop the "$\Delta$"s below to reduce clutter.

For a state $|\psi\rangle$, the Connes cocycle (CC) flow operator [43,44] is defined as

$$u_s\left(\psi|\omega;\overline{b}\right) = \rho_{\omega;\overline{b}}^{\mathrm{is}} \rho_{\psi;\overline{b}}^{-\mathrm{is}}. \tag{18}$$

This operator (appropriately dressed) will be the one we use in the main argument of section 4.2 below; notice that it is an operator in the bulk theory and does not make reference to the boundary dual. The reduced density matrix and one-sided modular Hamiltonian are not well-defined operators in the continuum, but the CC flow is a well-defined unitary. An amazing point is that it is possible to work out exactly the stress tensor in the CC-flowed-state

$$|\psi_s\rangle \equiv u_s\left(\psi|\omega;\overline{b}\right)|\psi\rangle. \tag{19}$$

Defining

$$t_s(w) \equiv \langle \psi_s | \Delta t(w) | \psi_s \rangle, \tag{20}$$

and similarly for $\bar{t}$, the expectation values are

$$t_s(w) = t_0(w)\theta(w_0 - w) + e^{-4\pi s} t_0 \left[ w e^{-2\pi s} \right] + \left( e^{-2\pi s} - 1 \right) \frac{\partial S}{2\pi} \delta(w - w_0),$$

$$\bar{t}_s(\bar{w}) = \bar{t}_0(\bar{w})\theta(\bar{w}_0 - \bar{w}) + e^{4\pi s} \bar{t}_0 \left[ \bar{w} e^{2\pi s} \right] + \left( e^{2\pi s} - 1 \right) \frac{\bar{\partial} S}{2\pi} \delta(\bar{w} - \bar{w}_0). \tag{21}$$

The derivation of this can be found in [45].[10] Colloquially, these equations say that the CC flow boosts all the energy in $\overline{b}$ and adds a pair of null shocks at $\eth \overline{b}$. It is most interesting to study the effect of the flow on the QHANE,

$$E_{QHA,s} \equiv 2\pi \int_{w_0}^{\infty} \mathrm{d}w \, t_s(w) + \partial S_{E,\psi}(w_0) = e^{-2\pi s} E_{QHA,0}, \tag{22}$$

using the fact that since $u_s$ is a one-sided unitary it doesn't change the entanglement entropy. Thus, the action of the CC flow can be summarised as 'boosting away the QHANE.'

---

[10]All the results mentioned in this section can be extended to multi-component regions [17]. Instead of using the vacuum, we use the split vacuum, which is the same as the vacuum in each component but differs in that the different components are factorised.

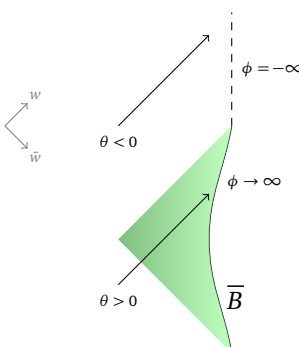

Figure 2: A null ray that hits the asymptotic boundary has positive expansion, and one that doesn't has negative expansion.

## 3.3 Focusing and CWI

CCWE is causal wedge inclusion for a class of bulk states. Causal wedge inclusion for a region $\bar{b}$ is the statement that $\eth\bar{b}$ is not causally related to $\overline{B}$. In other words, null rays shot out from $\eth\bar{b}$ to the future/past do not intersect $\overline{B}$. Take $\overline{B}$ to be the right boundary and $\bar{b}$ to be as in (10).

The expansions are defined as[11]

$$\theta(w,\bar{w}) \equiv \partial\phi(w,\bar{w}), \qquad \bar{\theta}(w,\bar{w}) \equiv \bar{\partial}\phi(w,\bar{w}). \tag{23}$$

Then, consider at the expansions at the end of the future-directed ray shot out from $\eth\bar{b}$, namely $\theta(\infty,\bar{w}_0)$. If this expansion is positive, then the light ray has hit the asymptotic boundary. If it is zero or negative, then it is to the future of the asymptotic boundary. Similarly, for a past-directed ray, we check the sign of $\bar{\theta}(w_0,\infty)$ to see if the ray hits the asymptotic boundary or lies to its past. This is shown in figure 2.

We can relate the expansions at $(w_0,\bar{w}_0)$ and $(\infty,\bar{w}_0)$ using the Raychaudhuri equation for JT gravity,

$$\partial\theta = \Gamma^w_{ww}\theta - 2\pi\langle t(w)\rangle_\psi. \tag{24}$$

The $SL(2,\mathbb{C})$ isometry can be used to set $w_0 = \bar{w}_0 = 0$, which also makes the Christoffel symbols vanish.[12] Integrating (24), we find

$$\theta(\infty,0) = \theta(0,0) - 2\pi\int_0^\infty dw\,\langle t(w)\rangle. \tag{25}$$

The implication of this for the quantum expansion,

$$\Theta \equiv \theta + \partial S_E(\psi), \tag{26}$$

is suggestive. Adding and subtracting $\partial S_E(\psi)$ to (25) gives

$$\theta(\infty,0) = \Theta(0,0) - E_{QHA}. \tag{27}$$

The fact that the second term is non-positive, due to the QHANEC, can also be considered a consequence of quantum focusing [46, 47].

---

[11]What is usually called expansion is more analogous to $\partial\phi/\phi$. But $\partial\phi$ is a simpler quantity for JT gravity.
[12]$\Gamma^w_{ww} = 2\bar{w}/(1-w\bar{w})$.

It is technically not correct to calculate the expansion at $w = \infty$, since $\mathcal{M}$ might end at a smaller value of $w$. This would be due to the boundary condition (5) being satisfied for $w_{\max} < \infty$. But that need not worry us, due to focusing. Specifically, if the spacetime ends at $w = w_{\max}$, $t(w > w_{\max}) = 0$, and

$$
\begin{aligned}
\partial \theta \leq 0 &\implies \theta(\infty, 0) \leq \theta(w_{\max}, 0), \\
\therefore \quad \theta(\infty, 0) \geq 0 &\implies \theta(w_{\max}, 0) \geq 0.
\end{aligned}
\tag{28}
$$

It is also common to define some $\phi_{\min} \leq 0$, such that $\phi = \phi_{\min}$ defines the 'singularity.' If a null ray hits this singularity surface at $w = w_{\max}$, then $\theta(w_{\max}, 0) < 0$ since for $w < w_{\max}$ the ray is inside the spacetime and therefore $\phi(w < w_{\max}, 0) > \phi(w_{\max}, 0)$. Again, due to focusing, $\theta(w > w_{\max}, 0) \leq \theta(w_{\max}, 0) < 0$.

So, the sign of $\theta(\infty, 0)$, which is also the sign of $\Theta(0, 0) - E_{QHA}$, is a good diagnostic for whether a future-directed rightwards null ray from $(0,0)$ hits the asymptotic boundary. Similarly, the sign of $\bar{\theta}(0, \infty)$ answers the same question for a past-directed rightwards null ray. If both of these are negative, the right boundary $\overline{B}$ is spacelike-separated from $\eth\overline{b}$ and $\overline{b}$ satisfies CWI.

## 3.4 How to properly dress a unitary

As a final step before the main argument, let me discuss subtleties of dressing. The main step in the argument below will be to use the backreaction of a bulk CFT unitary $u'$ localised within $\overline{b}$ to prove inconsistency of a particular claim with CCWE. However, the bulk CFT is not the full theory; to make $u'$ an operator within the JT+CFT system, we will need to dress the unitary $u'$ so that its support is within $\overline{b}$, even after taking into account backreaction. This can be subtle, and I will assume it below, see assumption A.4. In this section, I argue for the plausibility of this assumption.

The main subtlety is what was called 'gravitational spreading' in [17]. In general gravitational theories, the dressed operator will not be exactly the same as the QFT operator. To see why, expand $u'$ as a sum over products of bulk local operators $a_1(x)a_2(y) + \dots$. With gravitational dressing, $x, y$ cannot be specified as coordinates but have to be specified in a diffeomorphism-invariant way. Suppose that they are defined relationally to some point $P_0$, so it is actually $|x - P_0|, |y - P_0|$ that are fixed. But the action of the $a_2$ operator actually changes the geometry. Thus the distances $|y - P_0|$ and $|x - P_0|$ are calculated in slightly different geometries; the former is calculated in $g_\Psi$ and the latter in $g_{a_2\Psi}$. So the distance between $a_1$ and $a_2$ is slightly different due to backreaction. While we used a specific dressing to make this argument, it is easy to see that it is a much more general problem, which can be stated mathematically as a non-trivial commutator between $a_1$ and the dressing of $a_2$.

To define $u'$, then, we need a dressing that is *guaranteed* not to spread into $b$. Dressing it to $\overline{B}$ is not an option, because backreaction can reduce the (renormalised) distance between $\overline{B}$ and $\eth\overline{b}$. But dressing it to $\eth\overline{b} = \eth b$ is. This is because the local properties at $\eth\overline{b}$ are fixed from $b$.[13] Specifically, the dressing must require that every local operator making up $u'$ must be located to the right of $\eth\overline{b}$. This ensures that spreading does not cause the unitary to leak into $b$.

The reader might be worried that this dressing doesn't quite ensure that $u'$ commutes with $a_b$, however. For example, if $b$ (and therefore $\eth b$) is defined relationally to $B$, we have set up a chain where $u'$ is dressed to $\eth b$ which itself is dressed to $B$, and therefore that $u'$ doesn't commute with $a_b$ since $a_b$ doesn't commute with the dressing of $\eth b$. But such a definition of $b$ is a contradiction already, since $\phi(\eth b) = \phi(\eth\overline{b})$. Thus it must be that $\eth b$ is

---

[13]In an appropriately limiting sense, since operators localised to a point typically have divergent fluctuations. In the semi-classical limit these fluctuations should be suppressed in $\phi_r$.

gauge-invariantly defined in a way that does not make reference to any properties in the bulk of $b$ or $\bar{b}$. Extremality is one such definition, but "fixed renormalised distance from the right boundary" is not.[14]

Thus, it seems reasonable to dress $u'$ to $\eth b$. But there is another subtlety because spreading *does* deform the unitary $u'$, even while keeping it localised in $\bar{b}$. In calculating the backreaction of $u'$, we need to calculate the stress tensor in $u' |\Psi\rangle$; but in general spreading makes it hard to calculate this even if we know the stress tensor expectation value in $u' |\psi\rangle$. This effect can only be ignored for unitaries whose energy is small, precluding macroscopic rearrangements of the bulk stress-energy. An important advantage of JT gravity is that the metric of the bulk $AdS_2$ is fixed. Backreaction is entirely in the change of the dilaton profile and therefore the trajectories of the boundary particles. Since we work in the limit $\epsilon \to 0$ where the boundary particles are infinitely far away, the bulk CFT does not see this change [50], and so $\langle t \rangle_{u'\Psi} \approx \langle t \rangle_{u'\psi}$ (note the change of case in the subscript).

While I have argued here that dressing $u'$ to $\eth b$ is enough to prevent spreading into $b$, I will *not* assume that this is the dressing. All I assume is that there exists *some* gauge-invariant operator that agrees with $u'$ at $\phi_r \to \infty$ and does not spread into $b$, i.e. A.4. Further, I will not assume that this can be done for arbitrary bulk regions $\bar{b}$, but only for entanglement wedges; the importance of this will be highlighted in section 6.1.

## 4 The main argument

Suppose someone claims that $W_E[B; \Psi] = b$, such that $b \supseteq W_C[B; g_\Psi]$ and $\bar{b} \supseteq W_C[\overline{B}; g_\Psi]$. Call this the $Bb$ claim. I argue in section 4.2, based on assumptions listed in section 4.1, that most $Bb$ claims — those where $b$ is not quantum extremal — are false.

Take $B$ to be the left boundary, and $b$ to be the region $w < w_0, \bar{w} < \bar{w}_0$. If $\eth b$ is not a QES, then either $\Theta(w_0, \bar{w}_0) \neq 0$ or $\bar{\Theta}(w_0, \bar{w}_0) \neq 0$. The quantum expansion in one of the four directions $\pm w, \pm \bar{w}$ is positive, assuming that there are no divergences in $t_0, \bar{t}_0$ at $(w_0, \bar{w}_0)$.[15] Let's assume that $\Theta(w_0, \bar{w}_0) > 0$ for definiteness; the argument below is essentially unchanged if one of the other three quantum expansions is positive. As before, the $SL(2, \mathbb{C})$ isometry can be used to set $w_0 = \bar{w}_0 = 0$.

### 4.1 Assumptions

Apart from standard assumptions such as the existence of a single semi-classical bulk geometry, I will also use

A.1 Complementary Causal Wedge Exclusion (CCWE).

A.2 Complementary recovery,

$$\overline{W_E[B; \Psi]} = W_E[\overline{B}; \Psi]. \tag{29}$$

A.3 $W_E[B; \Psi]$ does not contain islands. In other words, $\eth W_E[B; \Psi]$ is a single point.

This assumption allows the use of the results of section 3.3 below.

A.4 The bulk CC flow unitary for $W_E[\overline{B}; \Psi]$ — $u_s(\psi|\omega; W_E[\overline{B}; \Psi])$ — can be dressed so that its support is entirely spacelike to $W_E[B; \Psi]$.

---

[14]Indeed, the fact that the reconstruction wedge can disagree with the entanglement wedge [48,49] is precisely a consequence of the fact that the property of being a *minimal* extremal surface is not a definition of this sort.

[15]All divergences can be smoothed out, so this is not a serious restriction.

See section 3.4 for further discussion of this assumption.

Let me pause here to discuss the epistemic status of these assumptions. Assumption A.1 is the condition discussed in section 2, and an assumption only to avoid a rather odd possibility as discussed above. Assumptions A.2 and A.3 limit the scope of the argument to certain cases; and this set of cases is expected to not be the empty set. The only non-trivial assumption, then, is A.4, and it is true a posteriori [17, 51, 52].

## 4.2 The central result: Bulk subregions need to be bounded by QESs

The argument begins with assuming the truth of the $Bb$-claim. Because of the $Bb$-claim and assumption A.2, we have that $\overline{b} = W_E[\overline{B}; \Psi]$. By assumption A.4, we have that the operator,

$$u'_s \equiv u_s\left(\psi | \omega; \overline{b}\right),\tag{30}$$

can be dressed such that its support remains spacelike to $b$.

To check CCWE, we need to calculate the bulk dual of $u'_s |\Psi\rangle$. Because $u'_s$ is localised within $\overline{b}$, it changes the geometry of $\overline{b}$ but leaves that of $b$ unchanged. Denote the wedge with the new geometry, including the backreaction of $u'_s$, by $\overline{b}_s$. To construct the geometry dual to $u'_s |\Psi\rangle$, we need to paste $\overline{b}_s$ to $\overline{b}$ at $\eth\overline{b}$, imposing the codimension-two junction conditions of [53]. The first condition is that the dilaton is continuous,

$$\lim_{w,\bar{w}\to 0^+} \phi_s(w,\bar{w}) = \lim_{w,\bar{w}\to 0^-} \phi(w,\bar{w}) = \phi(0,0).\tag{31}$$

The second condition is that the discontinuity of the expansion is given by the delta-function contribution in the null-null component of the stress tensor,[16]

$$\left[\theta\right]_{0^-}^{0^+} = \text{sing}\,\langle t(w=0)\rangle\,, \qquad \left[\bar{\theta}\right]_{0^-}^{0^+} = \text{sing}\,\langle\bar{t}(\bar{w}=0)\rangle\,.\tag{32}$$

These must be satisfied by $\overline{b}_s$. Integrating the Raychaudhuri equation (24) from $w=0^-$ — because $\Theta_s(0^-, 0) = \Theta_0(0,0)$ — to $\infty$ gives

$$\theta_s(\infty, 0) = \Theta_0(0,0) - e^{-2\pi s}E_{QHA} \xrightarrow{s>s_*} \text{positive},\tag{33}$$

where

$$s_* \equiv \frac{1}{2\pi}\log\frac{E_{QHA}}{\Theta_0(0,0)}\,.\tag{34}$$

By the discussion around figure 2, this means that for large enough $s$, a null ray shot out from $\eth\overline{b}$ hits the right boundary.[17] Thus, defining $s_1 = s_* + \delta$ with $\delta > 0$,

$$\Theta(\eth b) > 0 \quad\Longrightarrow\quad \overline{W_C[B; g_{u_{s_1}\Psi}]} \not\supseteq \overline{b} \xRightarrow{\text{CCWE}} \overline{b} \neq W_E[\overline{B}; \Psi].\tag{35}$$

There are two cases where the above contradiction doesn't arise. First, when $\Theta(0,0) = \bar{\Theta}(0,0) = 0$; in that case, there is no choice of $\overline{b}$ for which we can run the above logic and find $s_* < \infty$. But this is exactly the case when $\eth b$ is a QES. The other is when $E_{QHA} = 0$. But when the QHANE is 0, $\theta_0(\infty, 0) = \Theta_0(0,0)$ and so we must have $\Theta_0(0,0) \leq 0$ by CWI for $\overline{B}$. But $\Theta_0(0,0) \geq 0$ by assumption, and so we again have that $\eth b$ is a QES.

This completes the argument that $W_E[B; \Psi]$ must be a quantum extremal wedge.

---

[16]Here,

$$\text{sing}\,f(x=x_0) \equiv \lim_{\delta\to 0}\int_{x_0-\delta}^{x_0+\delta} \mathrm{d}x\, f(x)\,.$$

[17]For $s \gg s_*$, the other shock gets reflected off the boundary and its effect needs to be taken into account, as explained in [17]. Fortunately, CCWE is already violated when $s - s_*$ is positive but small, when we can ignore this effect.

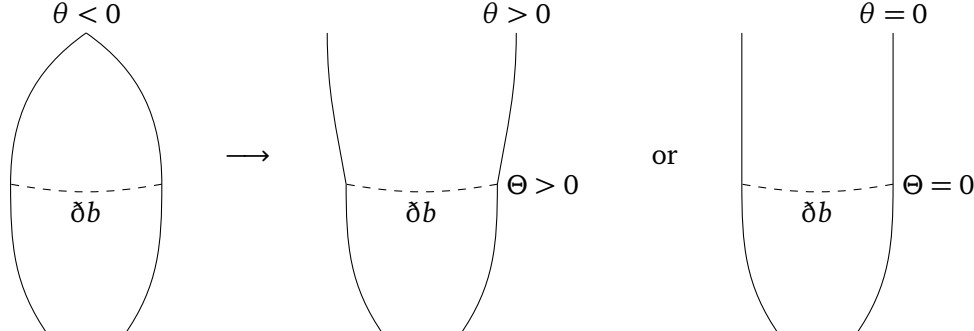

Figure 3: An illustration of the effect of the bulk CC flow on the null congruence passing through ð$b$. The dilaton has been visualised as the radius of an $S^1$. Left: in the original spacetime, the congruence collapses into the singularity. Middle: The case of positive quantum expansion at ð$b$. The negative energy shock causes a violation of classical focusing at ð$b$, and also boosts away the matter so that there is less focusing in the future. The congruence no longer collapses into a singularity but keeps expanding, eventually reaching the asymptotic boundary. Right: In the quantum extremal case, the congruence neither expands not contracts, eventually becoming the event horizon. More involved figures showing a similar lesson can be found in [17].

**Summary of the argument** The above argument implies that, if ð$b$ is not extremal, then the bulk CC flow unitary engineers the situation shown in figure 1, up to a possible renaming $B \leftrightarrow \overline{B}$. Thus, if $b$ were genuinely the bulk dual of $B$, then entanglement wedge reconstruction would contradict boundary causality: since $B$ and $\overline{B}$ are not in causal contact on the boundary, they should likewise not be in causal contact through the bulk.

The specific unitary we used, the bulk CC flow, can be thought of as one that de-focuses the null congruence to the future of ð$b$. In the spacetime $g_\Psi$, the congruence passing through ð$b$ focuses, due to matter flux; but in the spacetime $g_{u'\Psi}$, the matter flux is boosted away, reducing the focusing effect, and the congruence no longer collapses into a singularity. Instead, it reaches the asymptotic boundary. This is shown in figure 3, where the dilaton value is visualised as the size of an $S^1$.

The role of extremality is simple. The final expansion of the congruence is upper-bounded by the quantum expansion of ð$b$. If the final expansion is positive, it necessarily reaches the asymptotic boundary. It is only in the extremal case that the quantum expansion at ð$b$ is not positive in any of the four directions.

# 5 The case with islands

A similar logic works in the case where $b$ is allowed to contain islands, but with stronger assumptions. Define a geometric purification as a state $|\widetilde{\Psi}_B\rangle$ that purifies $\Psi$ reduced to $B$ and has a semi-classical bulk dual. One expects the bulk dual of this purification to contain a region with the same metric and reduced bulk state as $W_E[B; \Psi]$. I will assume the existence of a special type of geometric purification and run the previous argument again. This assumption is non-trivial and may be false.

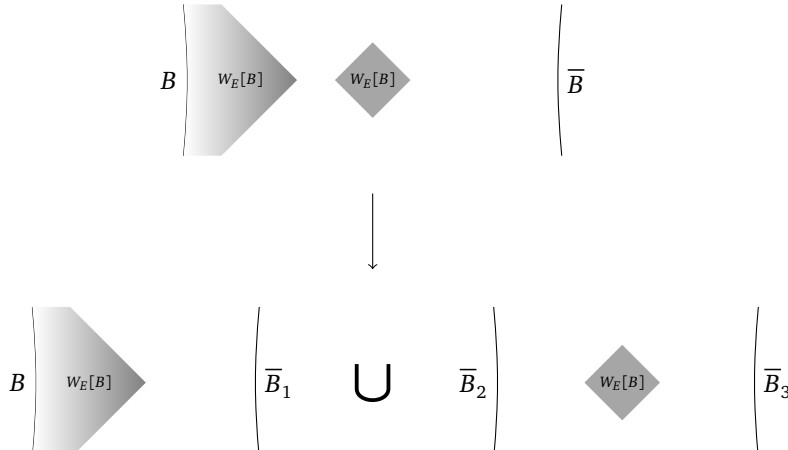

Figure 4: An illustration of assumption $A_I.2$. In the original spacetime, the entanglement wedge of $B$ has two components and three corners. In the new purification, each corner is homotopic to an asymptotic boundary.

## 5.1 Assumptions

Apart from the assumptions A.1, A.2 and A.4, I also need

$A_I.1$  The bulk CFT satisfies the split property [54].

$A_I.2$  There is a geometric purification $|\widetilde{\Psi}_B\rangle$ such that (a) it contains a region with the same metric and quantum state as $W_E[B;\Psi]$ and (b) every point in $\eth W_E[B;\Psi]$ is homotopic to an asymptotic boundary, as illustrated in figure 4.

$A_I.3$  If $b$ is the max-EW of $B$ in the state $|\Psi\rangle$, then it is also the max-EW in the purification assumed above.

## 5.2 Plausibility of assumption $A_I.2$

Suppose $W_E[B;\Psi]$ has $n$ connected components with bulk algebras $a_{1,...n}$. By the split property, there is a unitary $U_{\mathrm{sp}}$ from the CFT Hilbert space $\mathscr{H}_{\mathrm{bulk}} \to \mathscr{H}_{\mathrm{bulk}}^{\otimes n}$, such that $U_{\mathrm{sp}} a_r U_{\mathrm{sp}}^\dagger$ acts on the $r^{\mathrm{th}}$ copy.

Assumption $A_I.2$ then adds that the action of $U_{\mathrm{sp}}$ followed by unitaries in $\left(U_{\mathrm{sp}} a_1 \vee \cdots \vee a_n U_{\mathrm{sp}}^\dagger\right)'$ can create a solution of the JT gravity equations. Note that the equations are satisfied inside the images of $W_E[B;\Psi]$. Thus, there is roughly the right amount of freedom to make the $n$ copies of the bulk on-shell. This is why I believe this assumption to be reasonable.

There might be further constraints that this very quick analysis hasn't revealed however.

## 5.3 The argument

Suppose we are handed a $Bb$ claim for a bulk region $b$ that has many components. The set of corners $\eth b$ can be divided into three sets. The first, $\eth_+ b$, is the set of points such that one of the two outward expansions is positive. The second, $\eth_- b$, is the set where both the outward expansions are negative. The third, $\eth_0 b$, is the set consisting of quantum extremal points.

To disprove the $Bb$ claim, we need to argue that CCWE is violated whenever either or both of the first two sets are non-empty. The strategy is just to use assumption $A_I.2$ and the result

of section 4. First, if $|\eth_+ b| > 0$, then construct the purification $|\widetilde{\Psi}_B\rangle$, and then the previous argument shows a violation of CCWE in this new spacetime.

Denote by $\widetilde{\overline{b}}$ the complement of $b$ in $g_{\widetilde{\Psi}_B}$. $\widetilde{\overline{b}}$ is a union of $n = |\eth b|$ wedge-like regions on $n$ different copies of AdS$_2$, by definition. The flowed state we need is

$$\left|\widetilde{\psi_s}\right\rangle \equiv u_s\left(\widetilde{\psi}|\omega^{\otimes n};\widetilde{\overline{b}}\right)\left|\widetilde{\psi}\right\rangle . \tag{36}$$

Since the modular Hamiltonian of $|\omega\rangle^{\otimes n}$ is a sum over all the copies, we again find

$$-\partial_{w_0} S_{\text{rel}}(\widetilde{\psi_s}|\omega^{\otimes n};\widetilde{\overline{b}}) = e^{-2\pi s}\left[-\partial_{w_0} S_{\text{rel}}\left(\widetilde{\psi}|\omega^{\otimes n};\widetilde{\overline{b}}\right)\right], \tag{37}$$

where the derivative is with respect to the coordinates of any point in $\eth b$. The focusing argument in section 3.3 goes through and we find a violation of CCWE for every point in $\eth_+ b$. This implies that $b$ is not the max-EW of $B$ in this new state; by assumption $A_I$.3, it cannot be the max-EW in $\Psi$, disproving the $Bb$ claim.

Similarly, if $|\eth_- b| > 0$, use assumption A.2 to shift focus to $\overline{B}$ and construct the purification $\widetilde{\Psi}_{\overline{B}}$ satisfying assumption $A_I$.2 for $\overline{b}$. Then, CCWE is violated for $\overline{B}$ in this new geometry. Thus, the only $Bb$ claims that don't fall afoul of CCWE are the ones where $b$ is bounded by a QES.

## 6 Discussion

I have argued that there is a consistency condition that straightforwardly restricts entanglement wedges to those bounded by QESs, in JT gravity coupled to a CFT. Using this consistency condition requires incorporating backreaction into the definition of subregion duality, something that is most natural in the approximate error-correction models of [6, 28, 48, 55].

The important advantages of this new derivation are as follows. First, it is purely Lorentzian. Secondly, it never uses the replica trick or the gravitational path integral, even as an intermediate step. This is conceptually important, because the mysterious power of the gravitational path integral has been a subject of much confusion in recent times, see e.g. [56]. It is interesting to tease out how much of this mysterious power is present only in the path integral as opposed to Lorentzian low-energy gravity.

The fact that this technique also helps us constrain $Bb$ claims including islands is quite interesting in this regard, since it is a purely Lorentzian argument showing that islands can exist. To create the bulk purification assumed in assumption $A_I$.2, we presumably need exponential complexity. This likely ties in to the argument of [55] that semi-classical gravity is protected by complexity.

An important caveat here is that this new argument is not operational. This is because it uses the full Heisenberg state $u'|\psi\rangle$, rather than the history where $u'$ is applied as a process. So it doesn't correspond to something a bulk observer can do.

It would also be interesting to investigate the assumptions further, to see if any can be proven or disproven.

### 6.1 A different perspective on CCWE

Here is a different perspective on the CCWE condition. Define a Hermitian operator $a'$ supported in $W_C[\overline{B};\Psi]$, defined relationally with respect to $\overline{B}$. Similarly, also define another Hermitian operator $a$ supported in $b$, dressed in any way such that it commutes with all operators in $\overline{b}$. Assume also that these operators have small energy, and consequently small backreaction.[18]

---

[18]The choice of Hermitian operator rather than unitary is purely aesthetic. All of the below goes through with appropriate modifications for the unitary $e^{i\delta a}$.

Suppose now that I had assumed A.4 not for $W_E[\overline{B};\Psi]$ but for arbitrary regions $\overline{b}$. Then for the CC flow unitary localised in $\overline{b}$

$$[a, u'_s] = 0, \tag{38}$$

because they have explicitly been dressed to be spacelike-separated. The commutator of $a, a'$ in the state $u'|\Psi\rangle$ is

$$C_{\text{see}} = \langle\Psi|u'^\dagger_s[a',a]u'_s|\Psi\rangle = \langle\Psi|[u'^\dagger_s a'u'_s, a]|\Psi\rangle \approx {}_{g_\Psi}\langle\psi|[u'^\dagger_s a'u'_s, a]|\psi\rangle_{g_\Psi} = 0. \tag{39}$$

The first equality is a consequence of (38). The second equality is a consequence of the unitary generating a local flow: since both $a$ and $u'^\dagger_s a'u'_s$ have small backreaction, we can evaluate the commutator as a field theory commutator on $g_\Psi$. This last commutator manifestly vanishes due to the spacelike separation of the operators.

But the implication of a violation of CCWE is exactly that some commutators of this class are non-zero! Clearly, assumption A.4 can not be true for arbitrary regions; it must be that $u'_s$ cannot be appropriately dressed for non-extremal regions. It would be interesting to see this non-existence explicitly from gravitational dressing.

## 6.2 A new derivation of the QES formula?

Consider the set of regions $b_i$ that satisfy the CCWE check. There is one for every QES. The HRT formula for the entropy of $B$ requires us to prove two more things. First, that $S_E(B;\Psi)$ is given by the generalised entropy $\phi(\eth b_i) + S_E(b_i;\psi)$ of one of these regions. Secondly, this region is in fact the one with minimal generalised entropy.

Begin with the assumption that one of these regions $b_{i_0}$ is the entanglement wedge $W_E[B;\Psi]$. Assuming that the results of [24] apply, the modular Hamiltonian of $B$ satisfies

$$K_{\Psi;B} = \hat{A} + K_{\psi;b}, \tag{40}$$

where the first term is a central operator in $\mathscr{a}_{W_E[B;\Psi]}$.[19]

Since $\eth W_E[B;\Psi]$ is quantum extremal, a natural choice for the central operator is some function of the dilaton $f[\phi(\eth W_E[B;\Psi])]$. This is central because the boundary is fixed by the quantum extremality condition, and at leading order $\phi(\eth W_E[B;\Psi])$ can be calculated independent of the bulk state.

The results in [57–59] then can be used to fix the function $f(\phi) = \phi$, as follows. The state $\lim_{s\to\infty} u'_s|\Psi\rangle$ has the property that a null ray shot out from $\eth b$ hits the future boundary of $\overline{B}$, and that there is no stress-energy flux through the horizon. The works [57–59] establish an equality between $K_{\Psi;B}$ and the dilaton on the horizon for states with this property (and states perturbatively close by). But, since $u'_s$ commutes with $\hat{A}$, it must be the same operator in $K_{u'\Psi;B}$ and $K_{\Psi;B}$. This determines the area operator.

The final part is minimality. Two possibilities for proving this are as follows. First, we can try to extend the QMS theorem of [6] to the case with a non-trivial area operator. Second, we can use the simpler QEC models similar to [24] and *assume* that the QEC has sectors corresponding to each of the $b_i$ being subsystems of the code subspace. Assuming that the areas of each of these QESs has small fluctuations for simplicity,

$$|\Psi\rangle \approx \frac{1}{\sqrt{n}}\bigoplus_{i=1}^{n}|\chi_{\alpha_i}\rangle|\psi_{\alpha_i}\rangle. \tag{41}$$

---

[19]Likely, one needs to use the split property to find a type $I_\infty$ algebra contained in $b$ and then use the precise statements in [27].

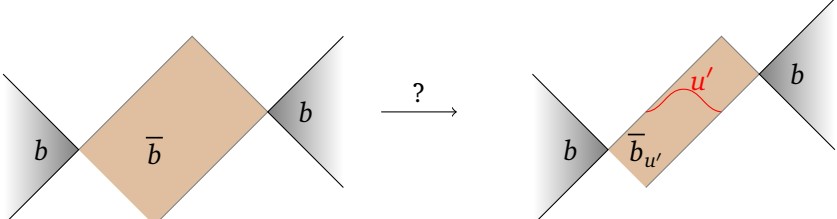

Figure 5: The action of a unitary in $\overline{b}$ should not change the spacelike-separated region $b$.

That all of these sectors have the same probability is an extra assumption here. Calculating the Rényi entropy for index $k$, we find that the minimal generalised entropy sector dominates. Making this precise would derive minimality for the entanglement entropy, and consequently also for subregion duality using [24].

### 6.3  Higher dimensions

There are many new issues to be dealt with in higher dimensions and other theories. It is likely that the argument involves a combination of the constructions in section 5 and [18]. Generalising this work to higher-derivative theories will also likely shed light on the relation between the entropies defined by Dong [60] and Wall [61].[20]

The output of the argument, assuming the myriad subtleties can be dealt with, should be that $\delta W_E[B]$ 'extremises the quantity that focuses.' In JT gravity, the quantity that focuses is the dilaton, see (24). [61] defined an entropy in higher-derivative theory as the quantity that focuses — i.e. by demanding the generalised second law — and noted a curious relation with the entropy defined using the replica trick in [60]. It would be interesting to flesh this out.

### 6.4  Defining gravitational subsystems

The original motivation for this work was the question of what constraints there are on gravitational subsystems. I have not come close to addressing this question, of course. First, let's outline the main obstacle to using the techniques here to address the more general question.

The question is as follows. Suppose someone hands you a gauge-invariantly defined subregion $b$ along with an approximate subalgebra within it and claims it is a subsystem in semiclassical gravity. For this to be true, a unitary $u'$ localised in a spacelike-supported region should commute with sufficiently simple local operators in $b$. Concretely, denote by $\overline{b}_{u'}$ the complementary region with the backreaction of $u'$ taken into account. The technical question is whether it is possible to paste $\overline{b}_{u'}$ to $b$ at $\delta b$, without violating diffeomorphism constraints.

The main difficulty with addressing this more general problem is the following. The constraints are first-order non-linear equations that we can integrate from a point inside $b$. So there is no problem at $\delta b$ itself; we can use the codimension-two junction conditions as initial data for the integration of the constraints into $\overline{b}_{u'}$. A contradiction can only be found if there is another "final condition" that is inconsistent with the initial conditions at $\delta b$. CCWE provides us with this "final condition," but it is harder to find one more generally.

One question is whether there should be any constraints at all. One reason to believe that there should be no constraints is that there have been a number of reasonable suggestions for local gravitational subsystems in recent times [59, 62–64]. It is not clear that they are in contradiction with there being constraints. [59, 64] explicitly work at leading order in $G_N$, and so don't take into account backreaction. [62, 63] provide a definition of entanglement wedges

---

[20]I thank Ayngaran Thavanesan, Diandian Wang, Zi-Yue Wang and Zihan Yan for discussions about this.

and entropies for arbitrary subregions, showing that their answer satisfies many properties that one can expect. They leave open the question of what algebra this is the entropy of, however.

## Acknowledgments

I thank Chris Akers, Raphael Bousso, Marine de Clerck, Jackson R. Fliss, Jonah Kudler-Flam, Adam Levine, Onkar Parrikar, Geoffrey Penington, Pratik Rath, Arvin Shahbazi-Moghaddam, Antony Speranza, Ayngaran Thavanesan, Aron Wall, Diandian Wang, Zi-Yue Wang, and Zihan Yan for discussions.

**Funding information** This work has been partially supported by STFC consolidated grant ST/T000694/1. I am supported by the Isaac Newton Trust grant "Quantum Cosmology and Emergent Time" and the (United States) Air Force Office of Scientific Research (AFOSR) grant "Tensor Networks and Holographic Spacetime".[21] I also thank UC Berkeley for hospitality.

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
