# Peer review of "Extremality as a Consistency Condition on Subregion Duality"

_SciPost Physics, doi:SciPost Phys. 17, 133 (2024)_

## Round 1 · Referee Report · Anonymous (Referee 1) · 2024-6-23

Strengths
1- very interesting result 2- clear exposition 3- clear study of assumptions
Report
This is a very nice paper that shows under some carefully studied assumptions that the entanglement wedge of a boundary subregion in JT coupled to a CFT must be bounded by a QES. By computing the Connes cocycle unitary of a boundary state with respect to the vacuum, the author is able to violate the property of complementary causal wedge exclusion in the case where the region is bounded by anything else than a QES, which leads to a contradiction.
I found the derivation in this paper interesting as it provides a very geometric and completely Lorentzian understanding of why a QES has special recovery properties. I also think the paper is careful with its assumptions and provides an extensive and honest analysis of where they might break down (especially in the case with islands). Therefore I am happy to recommend this paper for publication in SciPost.
I found the derivation in this paper interesting as it provides a very geometric and completely Lorentzian understanding of why a QES has special recovery properties. I also think the paper is careful with its assumptions and provides an extensive and honest analysis of where they might break down (especially in the case with islands). Therefore I am happy to recommend this paper for publication in SciPost.
Recommendation
Publish (easily meets expectations and criteria for this Journal; among top 50%)

Author: Ronak Soni on 2024-08-02 [id 4672]
(in reply to Report 3 on 2024-07-23)Warnings issued while processing user-supplied markup:
Add "#coerce:reST" or "#coerce:plain" as the first line of your text to force reStructuredText or no markup.
You may also contact the helpdesk if the formatting is incorrect and you are unable to edit your text.
Thank you for the review.
If some examples of problematic notation are clarified, that would go a long way in me being able to improve the writing.
As for the question, there has been a misunderstanding.
This argument never uses the bulk dual of the boundary CC flow. I only ever introduce the CC flow in the bulk EFT.
I will add further clarification on this point in the text.

---

## Round 1 · Referee Report · Anonymous (Referee 3) · 2024-7-23

Report
Based on the explicit computation in the context of JT gravity, the paper aims to demonstrate that the boundary of entanglement wedge being the Q.E.S follows naturally from the general self-consistency conditions of subregion dualities known as CCWE (complementary causal wedge exclusion). To make progress, a few additional assumptions were imposed regarding properties of the bulk dual in relation to CCWE. The paper is an interesting attempt to provide novel understanding of the emergence of entanglement wedge in the context of subregion dualities.
In my opinion, the paper in the current form possibly needs additional streamlining regarding the presentation, including the choice of notations etc. I do have the following naïve question regarding the argument, that I hope the author can address and explain in better terms. A crucial ingredient in the argument is to use the bulk dual of the connes-cocycle flow to construct counter-examples that would violate CCWE, if the bulk subregion is not bounded by the Q.E.S. It is unclear to me whether there is a possibility of cyclic logic here, since the known bulk dual of the connes-cocyle flow, i.e. in terms of bulk shock waves, is based on entanglement wedge reconstruction. Maybe I have missed some important steps here. Naively it seems that to avoid cyclic logic in the arguments, one needs an understanding of the bulk dual of the connes cocycle flow for the "off-shell” choice of bulk subregions — whatever that means. Some comments in this respect will be ideal for facilitating the understanding of the readers.
In my opinion, the paper in the current form possibly needs additional streamlining regarding the presentation, including the choice of notations etc. I do have the following naïve question regarding the argument, that I hope the author can address and explain in better terms. A crucial ingredient in the argument is to use the bulk dual of the connes-cocycle flow to construct counter-examples that would violate CCWE, if the bulk subregion is not bounded by the Q.E.S. It is unclear to me whether there is a possibility of cyclic logic here, since the known bulk dual of the connes-cocyle flow, i.e. in terms of bulk shock waves, is based on entanglement wedge reconstruction. Maybe I have missed some important steps here. Naively it seems that to avoid cyclic logic in the arguments, one needs an understanding of the bulk dual of the connes cocycle flow for the "off-shell” choice of bulk subregions — whatever that means. Some comments in this respect will be ideal for facilitating the understanding of the readers.
Recommendation
Ask for minor revision
Report
This paper argues that the entanglement wedge is bounded by the QES. Comparing with the old argument, the advantage of this paper is that it is purely in Lorentz picture. This paper studies an important question and provides inspiring discussions. Based on this reason, I suggest the paper to be published.
Recommendation
Publish (meets expectations and criteria for this Journal)

---

## Round 2 · Referee Report · Anonymous (Referee 4) · 2024-10-11

Strengths

A novel argument for QES formula.

Weaknesses

none

Report

This paper uses a nice idea of complementary causal wedge exclusion to argue that QES is the only option for the boundary of entanglement wedge. Explicit argument has been demonstrated in JT gravity.

Requested changes

The discussion around equation 4.7 which is the main argument of the paper is rather short. I would require an expanded paragraph to explain the logic in words. If this can be demonstrated by a figure that will be even better.

Recommendation

Publish (easily meets expectations and criteria for this Journal; among top 50%)

  • validity: high
  • significance: high
  • originality: high
  • clarity: high
  • formatting: excellent
  • grammar: excellent

Author:  Ronak Soni  on 2024-10-11  [id 4859]

(in reply to Report 1 on 2024-10-11)
Category:
remark

Thank you. This is a good suggestion and I will implement it.

---

## Round 2 · List of Changes

Rearranged the presentation for clarity.

Major changes: 1. More discussion of plausibility of assumptions (sec 4.1). 2. More steps in the logic of the main argument, including clear indication of where each assumption is used (sec 4.2). 3. Moved discussion of dressing to section 3.4 4. Changed some notation; now, all bulk operators are small letters.

---

## Editorial Decision

published